# Recent Advances in Co-Former Screening and Formation Prediction of Multicomponent Solid Forms of Low Molecular Weight Drugs

**DOI:** 10.3390/pharmaceutics15092174

**Published:** 2023-08-22

**Authors:** Yuehua Deng, Shiyuan Liu, Yanbin Jiang, Inês C. B. Martins, Thomas Rades

**Affiliations:** 1Guangdong Provincial Key Lab of Green Chemical Product Technology, School of Chemistry and Chemical Engineering, South China University of Technology, Guangzhou 510640, China; 202010104714@mail.scut.edu.cn (Y.D.); liushiyuan@scut.edu.cn (S.L.); 2Department of Pharmacy, University of Copenhagen, Universitetsparken 2, 2100 Copenhagen, Denmark; ines.martins@sund.ku.dk; 3School of Chemical Engineering, Guangdong University of Petrochemical Technology, Maoming 525000, China

**Keywords:** co-amorphous, co-crystal, co-former screening, formation prediction of multi-component solid forms

## Abstract

Multicomponent solid forms of low molecular weight drugs, such as co-crystals, salts, and co-amorphous systems, are a result of the combination of an active pharmaceutical ingredient (API) with a pharmaceutically acceptable co-former. These solid forms can enhance the physicochemical and pharmacokinetic properties of APIs, making them increasingly interesting and important in recent decades. Nevertheless, predicting the formation of API multicomponent solid forms in the early stages of formulation development can be challenging, as it often requires significant time and resources. To address this, empirical and computational methods have been developed to help screen for potential co-formers more efficiently and accurately, thus reducing the number of laboratory experiments needed. This review provides a comprehensive overview of current screening and prediction methods for the formation of API multicomponent solid forms, covering both crystalline states (co-crystals and salts) and amorphous forms (co-amorphous). Furthermore, it discusses recent advances and emerging trends in prediction methods, with a particular focus on artificial intelligence.

## 1. Introduction

Drug absorption after oral administration of active pharmaceutical ingredients (APIs) inter alia depends on their physicochemical properties (e.g., polymorphic form, drug solubility), their dose, and the local environment within the gastrointestinal tract. It has been reported that up to 90% of the currently developed APIs and about 40% of the approved drugs have poor biopharmaceutical properties as a consequence of their insufficient solubility in water [1,2]. The pharmaceutical industry is thus investing considerable efforts in searching for strategies to improve the solubility and bioavailability of APIs, without compromising their effectiveness. Several strategies [3], including nano- and micro-based drug delivery systems (e.g., extracellular vesicles [4], nanospheres and micelles [5], and nano- and microemulsions [6]), modified release solid dosage forms (e.g., in tablets and capsules [7]), and crystal engineering [8] (e.g., co-crystals [9] and co-amorphous forms) have been successfully investigated and partly implemented to improve the solubility and dissolution rate of APIs belonging to the biopharmaceutics classification system classes II and IV [10].

In 1989, Desiraju defined the term crystal engineering as “the comprehension of intermolecular interactions within crystal packing, and the use of this knowledge to engineer novel solid materials possessing specific physical and chemical characteristics” [11]. Crystal engineering is a powerful tool in the design of crystalline structures presenting different packing and intermolecular interactions. These crystalline structures are composed of one or more types of molecules (multicomponent solid forms). [12] Among these solid forms, co-crystals, salts, and co-amorphous systems (Figure 1) have been considered important in improving the physicochemical properties of APIs [13,14,15].

In 2018, the U.S. Food and Drug Administration (FDA) defined pharmaceutical co-crystals as crystalline materials composed of a neutral API and a second neutral molecule generally defined as a co-former (Figure 1). Co-crystals are formed in a stoichiometric ratio where the molecules interact via non-covalent interactions, such as hydrogen bonds, van der Waals interactions, π···π stacking, and halogen bonds, to form crystalline structures [13]. According to the FDA, this is the only difference between a co-crystal and a salt, as, in a pharmaceutical salt, a proton transfer occurs between an ionizable API and the co-former and both molecules interact in a stoichiometric way via charge-assisted hydrogen bond interactions (Figure 1) [16]. Solvates and hydrates are crystalline materials in which solvent molecules are present in the crystal lattice. Solvent molecules can thus be present in the crystalline lattice of co-crystals and salts, forming solvated/hydrated co-crystals and solvated/hydrated salts [16].

Amorphous solid dispersion technology has evolved since the 1960s, with polymers used as carriers. However, polymers have limitations, such as hygroscopicity and low drug loading. In 2009, Chieng et al. [17] introduced a term called “co-amorphous”, for amorphous solid dispersions that replace polymers with low molecular weight compounds. Co-amorphous refers to a homogenous, single-phase amorphous system containing two or more low molecular weight components interacting with each other in a non-periodic way, via, e.g., charge or non-charge assisted hydrogen bond interactions [18]. Based on the type of co-former used, co-amorphous systems can be divided into API–excipient and API–API co-amorphous systems [19]. In API–excipient co-amorphous systems, saccharin, nicotinamide, amino acids, dipeptides and carboxylic acids can be used as excipients. [20] On the other hand, in API–API co-amorphous systems, APIs can be selected according to their similar or complementary pharmacological properties [21].

Suitable co-formers are crucial to designing stable co-crystal, salts, and co-amorphous systems with desirable properties, such as high solubility, fast dissolution rate, good diffusion permeability, high physicochemical stability and a possible synergistic pharmacological effect [22]. Over the last few decades, high-throughput screening experiments have been used to predict the formation of API multicomponent solid forms. This trial and error approach is time consuming and requires a considerable consumption of materials and resources [23].

In recent years, advanced prediction methods, especially computer-assisted methods, have been developed to improve the specificity, sensitivity and accuracy of co-former screening and the formation prediction of API multicomponent solid forms. Figure 2 shows the timeline of the development of computer-assisted methods. Prediction methods for crystalline states were developed earlier (in 1997), some of which were explored for amorphous forms later (in 2011). Herein, methods are divided into two categories: hydrogen bond based methods and non-hydrogen bond based methods. The former category includes Δp*K*a based models [24,25,26,27,28]; supramolecular synthon engineering [29,30,31,32,33]; virtual co-crystal screening based on molecular electrostatic potential surfaces (MEPs) [34,35,36,37,38]; and hydrogen bond propensity (HBP) [39,40,41,42]. The latter category includes lattice energy calculation [43,44]; molecular complementarity (MC) by using the Cambridge Structural Database (CSD) [45,46,47]; the Hansen solubility parameter (HSP) [23,48,49,50]; conductor-like screening model for real solvents (COSMO-RS) [51,52,53,54,55]; artificial intelligence (AI) strategies [56,57,58,59,60]; and other novel methods [23,61,62]. Even though none of the methods can unfailingly predict the formation of API multicomponent solid forms, they can provide guidance for co-former screening to reduce the number of laboratory tests. The combination of two or more methods can greatly improve the effectiveness and accuracy of co-former screening [63,64,65]. Table 1 and Table 2 summarize cases in the recent literature using hydrogen bond based methods and non-hydrogen bond based methods, respectively.

## 2. Hydrogen Bond Based Methods

### 2.1. ΔpK_a_ Rule

The Δp*K*_a_ Rule is a simple approach to predict the formation of co-crystals or salts [63,68]. The difference in p*K*_a_ (Δp*K*_a_) between acidic/basic APIs and basic/acidic co-formers is calculated as follows:Δp*K*_a_ = p*K*_a_ (base) − p*K*_a_ (acid)(1)

Δp*K*_a_ can be used to estimate the tendency for proton transfer between a given API and co-former. When Δp*K*_a_ is higher than 3, a significant difference in the acidity or basicity of the API and co-former is obtained, indicating a preferable formation of a salt. When Δp*K*_a_ is lower than 0, the acidity or basicity of the API and co-former are similar and thus the formation of a co-crystal is expected where non-charge-assisted hydrogen bond interactions occur between the API and co-former. However, when Δp*K*_a_ is between 0 and 3, the difference in acidity or basicity is not large enough to clearly favor the formation of salt over a co-crystal or vice versa. In such cases, the Δp*K*_a_ rule is not the most reliable approach to predict the formation of a salt or co-crystal [9,69]. A combination of experimental and computational methods is thus necessary to determine the proton location. These methods include nuclear magnetic resonance spectroscopy, X-ray crystallography, and computational modeling using density functional theory or molecular dynamics simulations [70].

The Δp*K*_a_ rule was validated by Cruz-Cabeza using 6465 crystalline structures from the CSD [71]. Multicomponent crystalline solid forms with ionized or non-ionized acid-base pairs are only observed when Δp*K*_a_ is greater than 4 or less than -1, respectively. When Δp*K*_a_ is between -1 and 4, Δp*K*_a_ and the likelihood of proton transfer showed a linear relationship (Figure 3). A “salt–co-crystal continuum” exists when the proton position is ambiguous (Δp*K*_a_ between 0 and 3), which is between the two extremes of salt and co-crystal [63]. In such cases, the Δp*K*_a_ rule is not applicable for predicting the formation of salts or co-crystals. Childs et al. [63], investigated the propensity for forming co-crystals or salts with multicomponent pairs with Δp*K*_a_ lower than 3. They prepared a total of 20 multicomponent solid forms of theophylline and guest molecules, which included 13 co-crystals, five salts, and two within the salt–co-crystal continuum. It is important to note that the formation of salts or co-crystals is influenced by several factors, including the solvent used (and its polarity), temperature, API or co-former concentration, and crystal packing interactions. Therefore, the values of Δp*K*_a_ alone cannot predict with certainty the formation of salts or co-crystals, but they can provide useful information about the likelihood of their formation.

### 2.2. Supramolecular Synthons

Recognizing supramolecular interactions is crucial for designing crystals. After introducing the term “crystal engineering” in 1989, Desiraju [72] introduced the term “supramolecular synthons” and defined them as structural entities within supermolecules (i.e., complexes of two or more molecules that are non-covalently bonded) that can be created and/or arranged through intermolecular interactions using existing or feasible synthetic methods. Supramolecular synthons can serve as a design strategy for controlling the self-assembly of molecules through non-covalent interactions in the solid state. These synthons play a fundamental role in the formation of co-crystals because they act as building blocks in supramolecular chemistry, guiding the arrangement and organization of molecules in the solid state. Supramolecular synthons can be divided into two categories: homosynthons and heterosynthons (Figure 4). Supramolecular homosynthons consist of the same complementary functional groups that can form self-association motifs, such as dimers or chains, including acid-acid [73] and amide-amide [74] interactions. Supramolecular heterosynthons consist of diverse but complementary functional groups, including acid-amide [75], acid-hydroxyl [76], hydroxyl-pyridine [77], acid-pyridine [28], acid-N-oxide [70], amide-N-oxide [78], sulfonamide-N-oxide [79], and sulfonamide-amide [80] interactions. In a study conducted by the CSD, the hierarchy of supramolecular heterosynthons, involving carboxylic acids and alcohols, was evaluated in competitive environments, i.e., considering that both carboxylic acid and hydroxyl groups compete to form hydrogen bond motifs [28]. It was found that supramolecular heterosynthons (COOH···N_arom_ and OH···N_arom_) are preferred over the corresponding supramolecular homosynthons (COOH···COOH and OH···OH). Supramolecular synthons enable the prediction of the possible interactions between different molecules and the assessment of their propensity to form stable co-crystals. Through the analysis of complementary synthons in the molecular components, it becomes possible to narrow down the potential co-formers for a given target molecule.

The use of supramolecular synthons has been explored only in multicomponent crystal forms [31,32,70]. Since the supramolecular synthon principle is simple and does not require any calculations, it has been most frequently used as a tool to guide co-former selection. For example, when screening co-crystals of regorafenib (REG), it was found that REG has several hydrogen bond sites including hydrogen bond donors (amino groups), and hydrogen bond acceptors (carbonyl and pyridine groups) [31]. Dicarboxylic acids were chosen as co-former candidates because hydroxyl and carbonyl groups of carboxylic acids can act as hydrogen bond donor and acceptor, respectively. Three dicarboxylic acids including malonic acid, glutaric acid and pimelic acid were found to form co-crystals with REG. Based on the knowledge that nicotinamide and isonicotinamide commonly form heterosynthons with carboxylic acids, Das et al. [32] used nitrogen-containing carboxylic acids, including 3,5-pyrazole dicaboxylic acid, dipicolinic acid, or quinolinic acid as co-formers and successfully prepared new multicomponent crystal forms. Four different supramolecular heterosynthons were found in nicotinamide and isonicotinamide co-crystals with 3,5-pyrazole dicarboxylic acid (Figure 5). In co-former screening for minoxidil, Deng et al. [70] found that robust O-H···N or N-H···O hydrogen bonds were observed between pyrimidine N-oxide and carboxylic acids. Therefore, 17 co-former candidates with carboxylic acid groups were chosen and eight aromatic carboxylic acids successfully formed multicomponent crystal forms with minoxidil.

### 2.3. Molecular Electrostatic Potential Surfaces (MEPs)

The calculated MEPs generated using the density functional theory in the gas phase are used to identify surface site interaction points and predict electrostatic interactions at the surface of the molecules [81]. The strength of hydrogen bond donors and acceptors can be ranked according to the MEPs [82], which have also been further used in the design of ternary multicomponent crystal forms [83]. The hydrogen bond donor (*α*) and acceptor (*β*) are generated from the maxima and minima of the MEP and calculated using Equations (2) and (3), respectively.
(2)$$\alpha =0.0000162{\mathrm{MEP}}_{\max}{}^{2}+0.00962{\mathrm{MEP}}_{\max}$$
(3)$$\beta =0.000146{\mathrm{MEP}}_{\min}{}^{2}-0.00930{\mathrm{MEP}}_{\min}$$
where MEP_min_ and MEP_max_ are the local minima and maxima on the MEPs (unit: kJ·mol^−1^). The prediction of multicomponent crystal forms based on MEPs considers that all hydrogen bond sites on the surface of a molecule are not restricted by the internal molecular structure and are free to interact with other molecules in the solid-state environment. The molecular arrangement, steric constraints and packing effects are not taken into account [81]. The interaction site pairing energy *E* is the sum of all contacts across the surface of every molecule in the crystal, which can be calculated using Equation (4) [84].
(4)$$E=-\sum_{\mathrm{ij}}\alpha_{i}\beta_{j}$$
where *α*_i_ are hydrogen bond donor sites, and *β*_j_ are hydrogen bond acceptor sites. Positive or negative unpaired sites locate low electrostatic potential gaps or regions, making no contribution to the overall electrostatic interaction energy [81].

According to Etter, the first most positive *α*_i_ interacts with the first most negative *β*_j_, the second most positive *α*_i_ with the second most negative *β*_j_, and so forth until all of the interaction sites are covered [85]. Based on this theory, Musumeci et al. [81] proposed that the probability of forming a multicomponent crystal form can be predicted by the difference in the interaction site pairing energy (Δ*E*) between the total *E* of pure components and the *E* of the multicomponent crystal forms, using the following Equation (5).
Δ*E* = −(*E*_crys_ − n*E*_1_ − m*E*_2_) (5)
where *E*_crys_, *E*_1_ and *E*_2_ are the interaction site pairing energies of a multicomponent crystal of stoichiometry 1_n_2_m_, and the pure component solids, 1 and 2, respectively. The interaction site pairing energy of a multicomponent crystal is calculated in the same way as for a pure component solid. A high value of Δ*E* indicates a stronger interaction between two different components and a higher probability of forming a multicomponent crystal form. The minimum value of Δ*E* is 0 kJ·mol^−1^, indicating that the formation of multicomponent crystal forms must increase the interaction energy [81]. The specific cut-off value depends on the system involved.

The prediction reliability of MEPs has until now only been demonstrated for multicomponent crystal forms. For example, Pagliari et al. [86] applied MEP plots to illustrate the complementarity between amide sites and aromatic rings, which clarified how co-crystals are formed. Musumeci et al. [81] used caffeine and carbamazepine as model APIs and about 1000 co-former candidates to examine the reliability of the prediction method. The results demonstrated that when Δ*E* was over 11 kJ·mol^−1^, the possibility of co-crystal formation is larger than 50%. Grecu et al. [84] successfully validated MEPs as a virtual co-crystal screening tool using reported cases of 18 APIs. The co-former candidates were ranked according to their Δ*E* values and the ones presenting a larger value of Δ*E* were consistent with the experimental results in most cases. Furthermore, the MEPs method showed little difference when compared to the COSMO-based methods (see below). The MEPs method can also be an effective tool to (i) investigate the driving force of co-crystals/salts formation and explain the ratio of APIs and co-formers in multicomponent crystal forms [65,70], and (ii) elucidate the topology of hydrogen bonds and identify the factors contributing to any observed disorder in a crystal lattice [37]. Apart from hydrogen bond interaction, the MEPs method was also explored to give insight into halogen bond [87] and chalcogen bond [88] formation, which belongs to σ hole interactions and is coming into more focus recently. For instance, in co-crystals of 1,4-diiodotetrafluorobenzene and the isomeric *n*-pyridinealdazines (*n* = 2, 3 and 4), the I···N halogen bond interaction is the primary interaction [87]. The σ- and π-holes of the MEPs in 1,4-diiodotetrafluorobenzene showed high potential and were used as halogen bond donors. The halogen bond strength order for *n*-pyridinealdazines was 4 > 3 > 2 according to the *V*_s,min_ (minimum potentials) values located near the pyridine-N atom for *n* = 2, 3 and 4 (Figure 6). Wzgarda-Raj et al. [88] synthesized four new multicomponent crystals of trithiocyanuric acid with pyridine *N*-oxide derivatives, where N–H···S hydrogen bonds were observed to form R^2^_2_(8) synthons. Overall, the validated approach provides a promising framework for virtual multicomponent crystal forms screening and could potentially be applied in drug discovery.

### 2.4. Hydrogen Bond Propensity (HBP)

HBP was first developed by Galek et al. [89] as a logit (i.e., probabilistic) hydrogen bond propensity method to quantify the probability of hydrogen bond formation and used in the experimental polymorphic screening of ritonavir. This method was further used to successfully screen for multicomponent crystal forms of lamotrigine [90]. Based on the HBP model, it is important to recognize every potential hydrogen bond by identifying the donors and acceptors, which are responsible for the formation of multicomponent crystal forms.

HBP was developed by the Cambridge Crystallographic Data Centre and has been integrated into the Mercury program [91] to identify donors and acceptors forming usual or unusual hydrogen bonds. Molecules from the CSD with certain functional groups were used to prepare a dataset to train statistical models [41]. When predicting co-crystallization between an API and a co-former, both homomeric and heteromeric interactions are taken into account. If HBP_API-co-former_ is larger than HBP_API-API_ and HBP_co-former-co-former_, co-crystallization is likely to occur [92]. In Figure 7, A and B are two different molecules. A-A and B-B are homomeric interactions within A and B molecules, respectively. A-B and B-A are two different heteromeric interactions between A and B molecules. The Δ_propensity_ value is calculated as the difference of largest heteromeric interactions and largest homomeric interactions as shown in Equation (6).
Δ_propensity_ = (A-B)_best_ − (A-A)_best_(6)

The HBP method gives information about the possibility of forming a specific hydrogen bond, and it depends on how frequently the interaction occurs with respect to other interactions in a given fitting data set. In the HBP calculation for the system lenalidomide–nicotinamide [42], the -NH_2_ of nicotinamide group and the carbonyl group of lenalidomide were found to have a high propensity for heteromeric interactions (0.94), while homomeric interactions between lenalidomide–lenalidomide have a propensity of 0.84 for -NH- with carbonyl group, and 0.82 between nicotinamide–nicotinamide for -NH_2_ with carbonyl group (Figure 8). Heteromeric interactions were preferred as indicated by a Δ_propensity_ value of 0.1 and the co-crystal was successfully prepared experimentally. Majumde et al. [41] applied HBP in predicting indomethacin–nicotinamide (1:1) co-crystals, finding that their hydrogen bond motifs (N3-H···O5, N3-H···O4 and O3-H···N2), are the most likely donor–acceptor combinations predicted by HBP. Sarkar et al. [92] used HBP to predict the co-crystal formation between six model APIs and 25 possible co-formers. A success rate of 92–95% was obtained, indicating an excellent hydrogen bond affinity between APIs and co-formers. In another study, Sarkar et al. [93] used HBP, molecular complementarity, and hydrogen bond energy to predict co-crystal formation between six APIs and 42 potential co-formers. The combination of HBP and molecular complementarity allowed the achievement of an overall accuracy of 81%. Since HBP is a hydrogen bond-based method, its application is limited to multicomponent solid forms where hydrogen bonds are the dominant interaction [94].

## 3. Non-Hydrogen-Bond Based Methods

### 3.1. Lattice Energy

Lattice energy is another tool to predict multicomponent crystal forms. The energy difference between multicomponent crystal forms and pure individual components can be used as an indication of whether co-crystallization is expected to occur spontaneously, i.e., if it is thermodynamically favored compared to the crystallization of each starting material separately [95]. Considering that co-crystallization is commonly found to be a thermodynamically favorable process, its lattice energy is higher compared to the corresponding individual components [69,95,96]. The difference in lattice energies Δ*E*_latt_ can be calculated as follows:(7)$$\Delta E_{\mathrm{latt}}=E_{\mathrm{latt}}(A_{m}B_{n})-mE_{\mathrm{latt}}(A)-nE_{\mathrm{latt}}(B)$$
where *E*_latt_(A_m_B_n_) is the lattice energy of multicomponent crystal forms A_m_B_n_ consisting of molecules A and B in a stoichiometric ratio *m*:*n*, and *E*_latt_(A) and *E*_latt_(B) are the lattice energies of the pure components A and B, respectively. The probability of forming a multicomponent crystal form is higher when the Δ*E*_latt_ is more negative. It is unlikely that multicomponent crystal forms are formed if Δ*E*_latt_ has a positive value. Distinct from other methods, this prediction technique does not make any assumptions about hydrogen bonds and relies solely on the lattice energy. Factors such as temperature, pressure, solvent effects, and kinetics are not considered in the calculations [96].

The *E*_latt_ of the crystal can be calculated by three methods [97]: the first one is using a set of programs for crystallography, FlexCryst [98], to calculate free energy G. The second approach is using cohesive energy [99], and the last one is using total energy resulting from noncovalent, pairwise interactions between the molecule and its surrounding molecules [100]. Chan et al. [96] performed lattice energy calculations and investigated the thermodynamic stability of 102 multicomponent crystal forms containing nicotinamide, isonicotinamide, and picolinamide, 99 of which (more than 97%) were consistent with the observed tendency of the compound to crystallize. Kuleshova and co-workers [101] used the FlexCryst program suite to calculate the free energy and determine the relative stability of co-crystals of flavonoids and their pure crystal forms taken from the CSD. It was found that the lattice energy calculation was a valuable tool for in silico screening of co-crystal formation and stability, which can further be used to estimate their relative solubility. Sun and co-workers [102] proposed virtual co-former screening approaches, taking into account the lattice packing contributions of crystals to screen co-formers for indomethacin and paracetamol. If the lattice energy difference is ranked from smallest to largest, successful co-crystal formation can be found in the first six values of the lists (Figure 9). Lattice energy can also be used to determine the stable form of co-crystals. Surov et al. [103] compared the theoretical lattice energies and found that in multicomponent crystal forms of fluconazole with 4-hydroxybenzoic acid, hydrated crystal forms are more energetically favorable than the anhydrous co-crystals. Moreover, based on energy calculations, Vener and co-workers [97] found that the range of supramolecular synthons is approximately ~80 to ~30 kJ/mol, with a decreasing order of strength as follows: acid–amide > acid–pyridine > hydroxyl–acid > amide–amide > hydroxyl–pyridine.

### 3.2. Molecular Complementarity (MC, Fábián’s Method)

Molecular complementarity (MC) was first introduced by Fábián in 2009 [104] to investigate the molecular characteristics that affect the formation of multicomponent crystal forms. In his study, a statistical analysis was carried out using 131 descriptors of 1949 molecules. It was pointed out that molecules with similar properties, especially in molecular shape and polarity, tend to have a higher propensity to form stable multicomponent crystal forms. Five numerical descriptors including three shape descriptors (S-axis, S/L axis, and M/L axis) and two polarity descriptors (fraction of nitrogen and oxygen atoms, and dipole moment) were considered to be important in the formation of multicomponent crystal forms. S-axis is the length of the short axis, S/L axis is the short/long axis ratio, and M/L axis is the medium/long axis ratio, while L, M and S refer to the three unequal dimensions of a rectangular crystal cell, as shown in Figure 10 [93]. The MC method has been developed by the Cambridge Crystallographic Data Centre and incorporated into the Mercury software (version 3.7) [94]. Every descriptor has a criterion to indicate “PASS” or “FAIL”. A “PASS” indicates that a multicomponent crystal form was successfully formed, while a “FAIL” indicates that no multicomponent crystal forms were formed [104,105]. A multicomponent crystal form is likely to be formed only when all five descriptors demonstrate a “PASS.”

MC is mostly used as a preliminary screening tool instead of being used solely to determine the formation of multicomponent crystal forms. For example, Li et al. [106] used both COSMO-RS and MC in screening the formation of new multicomponent crystal forms of 2,4-dichlorophenoxyacetic acid. In total, 25 out of the 53 co-former candidates were identified to be potential co-formers, among which 20 co-formers were experimentally successful in forming new multicomponent crystal forms with 2,4-dichlorophenoxyacetic acid (Figure 11). Wu et al. [65] evaluated COSMO-RS, MC, HSP and their combinations in screening multicomponent crystal forms of 2-amino-4,6-dimethoxypyrimidine with 63 components. The overall successful rate of MC was 69.8%, and the best outcomes were obtained when using MC and COSMO-RS, with an overall success rate of 85.7%. Wu et al. [64] also used a combined method (COSMO-RS and MC) to screen multicomponent crystal forms for a pesticide, pymetrozine, with 39 co-former candidates. The calculation resulted in 13 promising co-formers, and it was discovered that nine of them produced novel solid phases.

### 3.3. Hansen Solubility Parameter (HSP)

The solubility parameter theory was proposed by Hildebrand and Scott in 1950 [107]. According to this theory, the cohesive energy indicates the total interactions in a material, including hydrogen bonds, van der Waals forces, covalent bonds and ionic bonds. Cohesive energy per unit volume, i.e., the cohesive energy density (CED), is an essential parameter used in pharmaceutical research to predict the physical and chemical properties of drugs, excipients and carriers (e.g., miscibility of a drug with excipients and carriers in solid dispersions). The relationship between the CED and the solubility parameter (*δ*, unit: MPa^0.5^) is shown in the following equation:(8)$$\delta ={\left(\mathrm{CED}\right)}^{0.5}={\left(\frac{\Delta E_{v}}{V_{m}}\right)}^{0.5}$$
where Δ*E*_v_ is the energy of vaporization, and *V*_m_ is the molar volume. The mutual solubility of two components is determined by their closeness in *δ* values. Desai and Patravale [108] applied the Hildebrand solubility parameter as one of the useful molecular descriptors to select co-formers for the formation of co-crystals with curcumin.

The Hildebrand and Scott approach is based on regular solution theory and works best for non-polar molecules interacting via weak dispersion forces [109]. In order to extend the application of this theory to more polar and strongly interacting systems, such as APIs, the HSP approach was developed by Hansen and divides the total solubility parameter (*δ*_t_), or Hildebrand solubility parameter, into three partial solubility parameters, as follows [110]:(9)$$\delta_{t}^{2}=\delta_{d}^{2}+\delta_{p}^{2}+\delta_{h}^{2}$$
where *δ*_d_ are dispersion forces (atomic dispersion), *δ*_p_ are ‘polar’ interactions, and *δ*_h_ are hydrogen bonds. These partial solubility parameters are most commonly calculated based on the Hoftyzer–Van Krevelen and Fedors group contribution methods, as follows: [111]
(10)δd=∑iFdi∑iVi,δp=∑iFpi20.5∑iVi,δh=∑iFhi∑iVi0.5
where *i* is the structural group within the molecule, *F*_di_, *F*_pi_, *F*_hi_ and *V*_i_ are the group contributions to the dispersion forces, polar forces, hydrogen bonding energy and molar volume, respectively.

In 1999, Greenhalgh et al. [112] proposed employing the difference of the total solubility parameter (Δ*δ*_t_) (Equation (11)) to predict the miscibility between the drug and carriers, and Mohammad et al. [48] proposed a cut-off value of Δ*δ*_t_ < 7 MPa^0.5^.
(11)$$\Delta \delta_{t}=\left|\delta_{t2}\right.-\left.\delta_{t1}\right|$$

According to Bagley et al. [113], *δ*_d_ and *δ*_p_ have thermodynamically similar effects, while *δ*_h_ has a different effect in nature [114], because they are interactions between hydrogen atoms and electronegative atoms such as O, N, and F. So, *δ*_d_ and *δ*_p_ parameters were combined as a volume-dependent solubility parameter, and *δ*_v_, and *R*_a(v)_ factor were introduced as follows:(12)$$\delta_{v}={\left(\delta_{d}{}^{2}+\delta_{p}{}^{2}\right)}^{0.5}$$
(13)$$R_{a\left(v\right)}={\left[4{\left(\delta_{v2}-\delta_{v1}\right)}^{2}+{\left(\delta_{h2}-\delta_{h1}\right)}^{2}\right]}^{0.5}$$

The two-dimensional plot of *δ*_v_ against *δ*_h_, i.e., the Bagley diagram, has been applied in different aspects, such as predicting the miscibility of two materials and the duration of drug intestinal absorption [115,116]. Albers et al. [117] found that for API and polymer systems, miscibility can be achieved if *R*_a(v)_ is not higher than 5.6 Mpa^0.5^.

Van Krevelen and Hoftyzer [118] introduced a three-dimensional approach to measure the difference in solubility parameters using Equation (14).
(14)$$\Delta \bar{\delta }={\left[{\left(\Delta \delta_{d}\right)}^{2}+{\left(\Delta \delta_{p}\right)}^{2}+{\left(\Delta \delta_{h}\right)}^{2}\right]}^{0.5}$$

Three parameters representing hydrogen bonding, polarity, and dispersion forces can be considered as three-dimensional coordinates of HSP space points. The two components are miscible if $\Delta \bar{\delta }$ ≤ 5 MPa^0.5^. For the purpose of convenient visualization of the spherical, instead of using the ellipsoidal solubility plot, a modified radius (*R*_a_) equation (Equation (15)) was proposed [119] as a representation of the Euclidean distance between the centrum HSP_1_ (*δ*_d1_, *δ*_p1_, *δ*_h1_) and another point HSP_2_ (*δ*_d2_, *δ*_p2_, *δ*_h2_) in the Hansen space (Figure 12).
(15)$$R_{a}={\left[{\left(4\Delta \delta_{d}\right)}^{2}+{\left(\Delta \delta_{p}\right)}^{2}+{\left(\Delta \delta_{h}\right)}^{2}\right]}^{0.5}$$

The HSP method was first used in co-crystal formation prediction in 2011 [48], and later, in 2020, it was used to indicate the good miscibility of tadalafil and repaglinide in co-amorphous systems [50]. The reliability of the HSP as a formation prediction tool for both co-crystal/salts and co-amorphous forms has been shown in a number of articles. For example, the HSP method was used to predict the formation of multicomponent crystal forms of minoxidil (MIN) [70] with 18 co-formers, and of 2-amino-4,6-dimethoxypyrimidine (MOP) [65] with 63 co-formers using the *R*_a(v)_, $\Delta \bar{\delta }$ and Δ*δ*_t_ criteria. Since there were large deviations from the experimental results for *R*_a(v)_ and $\Delta \bar{\delta }$ criteria, only the Δ*δ*_t_ criterion was chosen in further discussion in both MIN and MOP cases. The overall success rates of co-former prediction by the Δ*δ*_t_ criterion were 65.5% and 62.5% for MIN and MOP, respectively. In co-amorphous systems prediction, the Δ*δ*_t_ value between florfenicol and oxymatrine was determined to be 3.87 MPa^0.5^, indicating good miscibility between the API and co-former, contributing to the successful formation of a co-amorphous system [66]. The HSP method with the van Krevelen criterion $\Delta \bar{\delta }$ was successful in predicting four out of six norfloxacin co-amorphous systems. The results showed better accuracy than the Greenhalgh criterion Δ*δ*_t_, as it could offer more accurate information about the forces present between the molecules (e.g., polar interactions, hydrogen bonding and dispersion interactions) [67].

### 3.4. Conductor-like Screening Model for Real Solvents (COSMO-RS)

The COSMO-RS theory was developed by Klamt and co-workers [121] to predict the thermodynamic equilibria of pure components and liquid mixtures using static thermodynamic methods based on quantum chemical calculations. According to COSMO, the solute molecule can be represented as a collection of partial charges and the surrounding solvent can be approximated as a dielectric continuum of permittivity [122]. On the basis of COSMO, combined with statistical mechanics methods, Klamt [121,121] developed COSMO-RS to overcome some shortcomings of dielectric continuum solvation models. The COSMO-RS theory takes into account the molecular interactions (i.e., electrostatics, hydrogen bonding and Van der Waals interactions [123]) in calculating the chemical potentials of pure components and liquid mixtures.

COSMO-RS theory has been used in several research fields, including the prediction of solubility [122] and p*K*a [124], identification of suitable solvents [125], solvate or co-crystal formers [122], and the calculation of partitioning coefficients [126]. It is also a possible method for screening of multicomponent crystal forms [65,70]. Even though COSMO-RS theory has the capability of theoretically predicting the formation of a co-crystal or a co-amorphous system [51], the reliability of this method as a formation prediction tool only has been recently shown for multicomponent crystal forms [64,70]. On the basis of the assumption that the interactions between the API and co-former in multicomponent crystal forms are comparable to those in a virtual supercooled liquid, the formation of multicomponent crystal forms can be explained by liquid phase thermodynamics without considering the long-range packing order [69]. Therefore, the interaction strength of the API and co-former in the supercooled liquid state is able to predict co-crystallization, which is estimated using the excess enthalpy Δ*H*_ex_ (Equation (16)) of the API and co-former with a given stoichiometry as compared with the pure materials [127].
(16)$$\Delta H_{\mathrm{ex}}=H_{\mathrm{AB}}-x_{A}H_{A}-x_{B}H_{B}$$
where *H*_AB_ is the enthalpy of the supercooled stoichiometric mixture of components A and B. *H*_A_ and *H*_B_ are the enthalpy of the pure components A and B, respectively, and *x* is the mole fraction of each component. The probability of forming a multicomponent crystal form is higher when the Δ*H*_ex_ is more negative.

Although the COSMO-RS method ignores the solid-state order, it could successfully predict the formation of 21 new multicomponent crystal forms of 2-amino-4,6-dimethoxypyrimidine [65] with 63 co-former candidates with an overall success rate of 84.1% when Δ*H*_ex_ < −1 kcal/mol. Surprisingly, the COSMO-RS method showed up to 100% overall success rate in predicting the formation of multicomponent crystal forms of pymetrozine [64] and minoxidil [70] when Δ*H*_ex_ < −3.0 kcal·mol^−1^ and Δ*H*_ex_ < −2.00 kcal·mol^−1^, respectively. Alhadid et al. [128] calculated a solid–liquid phase diagram of 5 l-menthol/xylenol eutectic systems, where menthol/3,4-xylenol at 1:2 and 2:1 ratios, and l-menthol/3,5-xylenol at 1:1 ratio could form co-crystals. The authors further investigated solid-liquid equilibria data of the l-menthol/phenol eutectic system where two co-crystals were formed in 1:2 and 2:1 ratios, and successfully used the non-random two-liquid model (an activity coefficient model frequently applied to calculate phase equilibria) and COSMO-RS models to obtain the phase diagrams [129]. When exploring the possibility of forming new posaconazole co-crystals, COSMOquick selected 28 candidates from a list of approximately 140 potential co-formers. There were 13 new posaconazole multicomponent crystal forms (seven anhydrous, five hydrates, and one solvate) obtained [53]. The COSMO-RS method could also detect the transferability of co-crystallization between analogous molecules. Przybyłek et al. [52] used mixing enthalpy to evaluate the co-crystal formation of two methylxanthine derivatives, theophylline and caffeine, as shown in Figure 13. The mixing enthalpy of theophylline and caffeine showed a linear relationship and phenolic acids were found to be the most promising co-formers.

### 3.5. Artificial Intelligence (AI)

In recent years, AI methods, especially machine learning (ML), have emerged as promising and effective strategies to predict both multicomponent crystal forms and co-amorphous systems [130]. This method is mainly based on molecular descriptors representing the physicochemical properties of a molecule which are computed according to its chemical structure. Figure 14 displays the ML approaches commonly used. Building data sets based on experimental data is the first step in ML [131]. Molecules are characterized using artificial molecular features (such as molecular fingerprints and descriptors) or automatically extracted features (e.g., convolutional neural networks). ML algorithms are used to learn how a specific system is formed and include the creation and evaluation of models. The combination of AI with complementary experimental studies allows the leverage of the computational capabilities of AI to analyze vast amounts of data, identify patterns, and generate predictive models. Moreover, when AI predictions are complemented by experimental validations, it is possible to validate and refine these models, leading to a more comprehensive understanding of the underlying mechanisms of the formation of multicomponent solid forms. This may significantly reduce the time and cost of developing effective drugs [131], by creating a cluster analysis for co-former screening [132] and predicting the formation of multicomponent solid forms based on principal component analysis [133] and multivariate adaptive regression splines [134,135].

The main limitation in applying deep learning (DL) approaches in crystal engineering lies in the critical bottleneck of data availability, while the model performance also heavily relies on the key factor of data quality. The CSD comprises an extensive repository of high-quality positive samples (i.e., examples of successful multicomponent solid state form formation), providing valuable support for DL applications [130]. In data collection and augmentation, ML methods retrieve positive sample data from the CSD. Negative samples, however, are collected from experiments or experimental reports [130], or by computational methods such as network-based link prediction [135] and molecular similarity-based methods [57]. These data can be used to train ML models. Gröls et al. [56] developed different ML algorithms based on 1000 co-crystallization events and 2083 chemical descriptors to accurately predict mechanochemical co-crystallizations. The eXtreme Gradient Boosting model was able to identify three new co-crystals of diclofenac through mechanochemistry. Similarly, they developed and compared different machine learning and deep learning algorithms based on 418 experimental amorphization cases and 2066 molecular descriptors to predict mechanochemical amorphization [59]. The gradient boost model showed an accuracy of over 73% and was able to identify six novel co-amorphous systems with folic acid. Jiang et al. [130] used 6819 positive and 1052 negative samples to develop a graph neural network based on a deep learning model, as shown in Figure 15. The effectiveness of the graph neural network was shown by seven competitive models and three different and challenging out-of-sample tests, achieving prediction accuracy of higher than 96%. Wang et al. [57] developed a co-crystal prediction model based on the random forest approach. The positive samples were from the CSD and the negative ones were created by randomly combining different molecules into chemical pairs. The effectiveness of the model was shown by the formation of two co-crystals of captopril with l-proline and sarcosine. Przybyłek et al. [134] developed a MARSplines algorithm model containing 1D and 2D molecular descriptors and validated it for phenolic acid co-crystals. In a study of Meng-Lund et al. [62], molecular descriptors were computed for six APIs (carvedilol, mebendazole, carbamazepine, furosemide, indomethacin, and simvastatin), as well as for 20 naturally occurring amino acids, to build a partial least squares discriminant analysis (PLS-DA) model. Although this model was capable of accurately predicting 19 out of 20 mebendazole–amino acid co-amorphous combinations, it was limited to the use of amino acids as co-formers.

Chambers et al. [136] later built a model based on multivariate PLS-DA using the same six drugs and 20 amino acids. The model was then used to predict co-amorphous systems of mebendazole with 29 non-amino acid co-formers where 26 (90%) were correctly predicted. Deng et al. [67] calculated Pearson correlation coefficients and Ridge regressions between several molecular descriptors and co-amorphous forms. It was found that in norfloxacin co-amorphous salt systems, the formation of co-amorphous forms was dependent on the selection of a co-former with more π-conjugated rings, better miscibility with a smaller $\Delta \bar{\delta }$ value (ideally less than or close to 5 MPa^0.5^), and a larger Δp*K*a. Wang et al. [61] found that the aminopyridine synthon cannot completely guarantee the co-crystal formation between nimesulide and a series of pyridine analogues. By performing quantitative analysis of Ridge and Lasso regressions, it was discovered that the formation of co-crystals was influenced by several molecular descriptors of the co-formers, ranked in order of their impact as follows: MEPs, h_ema (sum of hydrogen bond acceptor strengths), Kier flex (molecular flexibility), and the horizontal distance of two N atom projections.

### 3.6. Other Approaches

ter Horst et al. [137] proposed a thermodynamic principle-based experimental method to discover co-crystals according to the solubility of their pure components (i.e., starting materials). The method considers the solubilities of individual components to identify suitable concentration ranges for exploring and discovering potential co-crystals, rather than focusing solely on the specific ratios or proportions of the co-crystal components. The screening steps are shown in Figure 16. Saturation temperature and reference temperature are terms determined from the phase diagram of a co-crystal system at different temperatures in solution. Co-crystals are likely to be formed if the saturation temperature is higher than the reference temperature by >10 °C. This method successfully led to the discovery of new co-crystals, including those involving carbamazepine with isonicotinamide, benzamide, and 3-nitrobenzamide, as well as cinnamic acid with 3-nitrobenzamide.

In recent years, computational methods have also been developed to screen co-formers more accurately and efficiently. A modified, non-bonded interaction energy model was introduced by Deng and co-workers [23] to predict the formation of co-amorphous systems. The formation of co-amorphous systems was believed to be driven by the interactions between the API and co-former. Their hypothesis suggests that the strength of these interactions directly influences the likelihood of co-amorphous formation. To quantify these molecular interactions, researchers calculate and denote the non-bonded energy difference (Δ*E*_non-bonded_) between the API and co-former. A higher absolute value of Δ*E*_non-bonded_ was indicative of a greater probability of co-amorphous formation. This model was developed based on 105 solvent-based cases involving 13 different drugs, and successfully predicted five new co-amorphous combinations of gefitinib and four new co-amorphous combinations of erlotinib.

## 4. Conclusions

Co-crystals, salts and co-amorphous systems are multicomponent solid forms containing one or more types of molecules. They are able to offer significant benefits to the bioavailability, solubility, dissolution, stability, permeability, and other physicochemical and pharmacokinetic properties of APIs. Choosing appropriate co-formers for a specific API is essential to develop solid formulations with ideal properties, which generally requires a significant amount of time and resources. Over the past few years, several knowledge-based methodologies, as well as computer-assisted prediction methods, have emerged exhibiting efficacy and accuracy in co-former selection and prediction of the formation of multicomponent solid forms. More strategies have been developed for multicomponent crystalline forms than amorphous forms. Some methodologies, such as HSP, and other new approaches, such as artificial intelligence, are applicable for both solid-state forms. Though none of the single approaches can completely and accurately predict the formation of multicomponent solid forms, each approach can offer valuable guidance in selecting the most suitable co-formers. The implementation of multiple screening/prediction approaches can result in considerable enhancements in the effectiveness and accuracy of predicting the formation of co-crystals, salts and co-amorphous systems, as confirmed by previous studies.

It can be expected that in the future, more investigations will continue to promote screening/prediction efficiency by using multiple approaches as well as developing novel models. One direction could be the integration and combination of various approaches mentioned in this review. For instance, combining machine learning algorithms with molecular docking methods could hold potential to improve the precision of multicomponent solid-state forms (co-crystal and co-amorphous) formations. The combination of quantum mechanical simulations with virtual screening techniques offers opportunities to delve deeper into the intricate interplay of thermodynamic stability and intermolecular interactions of multicomponent systems. High-throughput virtual screening methods taking into account advancements in computing power coupled with experimental screening to validate the predictions will speed up the identification of potential co-crystals and co-amorphous forms.

Collaborations between computational scientists and experimentalists are required to arrive at synergistic approaches to co-crystal and co-amorphous discovery. As researchers explore novel combinations of approaches, significant advancements in the field are foreseeable. These collective endeavors are poised to pave the way for the expedited discovery and development of innovative drug formulations, which could have a transformative impact on the pharmaceutical industry.

## Figures and Tables

**Figure 1 pharmaceutics-15-02174-f001:**
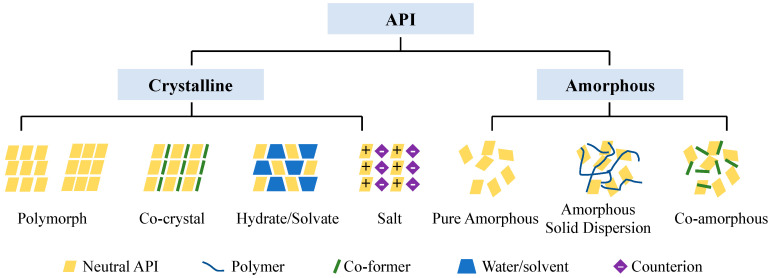
Representation of the different API solid-state forms.

**Figure 2 pharmaceutics-15-02174-f002:**
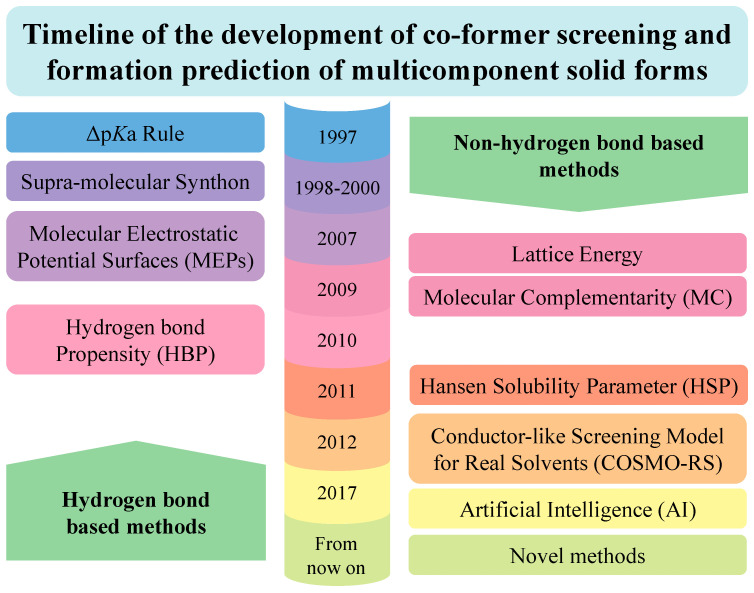
Timeline of the development of co-former screening and formation prediction of multicomponent solid forms.

**Figure 3 pharmaceutics-15-02174-f003:**
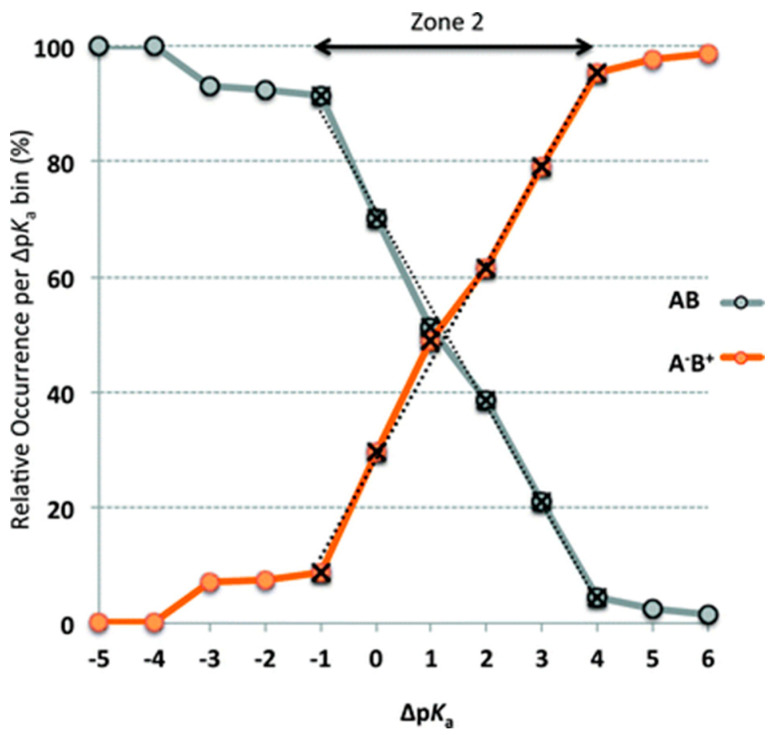
Relationship between the relative occurrence of co-crystals (AB neutral (grey)) and salts (A^−^B^+^ ionic (orange)) and the calculated Δp*K*_a_. Reprinted with permission from ref. [71]. Copyright 2012 Royal Society of Chemistry.

**Figure 4 pharmaceutics-15-02174-f004:**
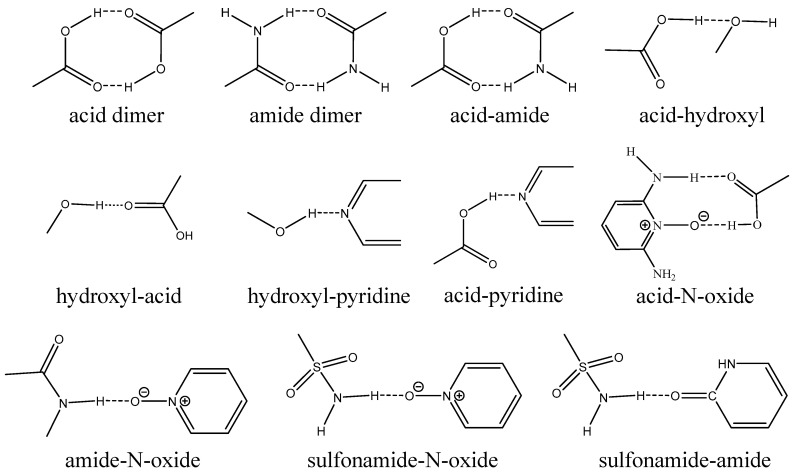
Different types of supramolecular synthons.

**Figure 5 pharmaceutics-15-02174-f005:**
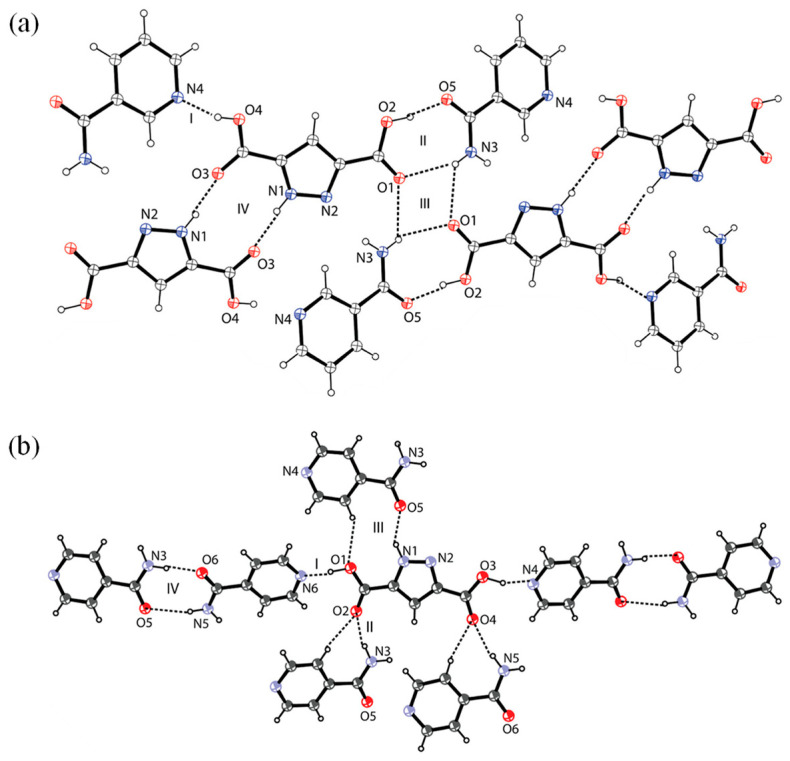
Four different synthons present in nicotinamide (**a**) and isonicotinamide (**b**) co-crystals with 3,5-pyrazole dicarboxylic acid. Reprinted with permission from ref. [32]. Copyright 2011 American Chemical Society.

**Figure 6 pharmaceutics-15-02174-f006:**
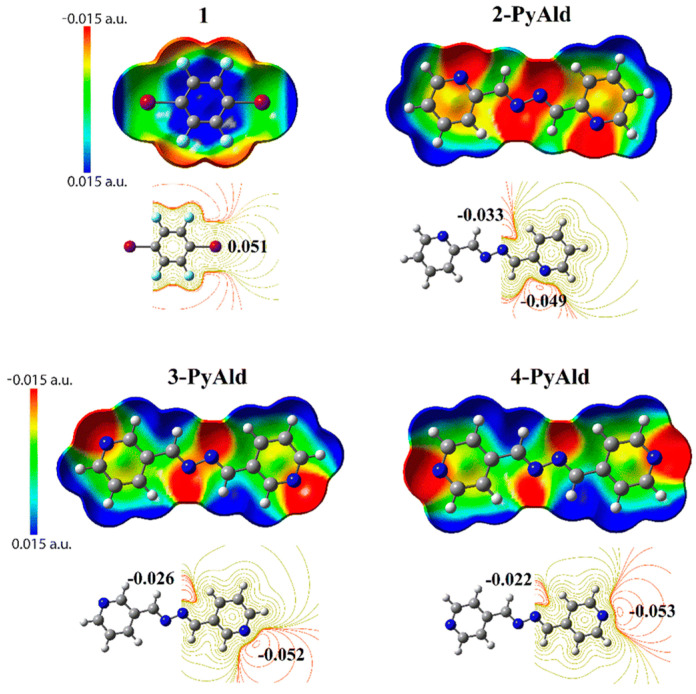
MEPs representation displaying charge distribution (red for charge concentration and blue for charge depletion). Reprinted with permission from ref. [87]. Copyright 2022 Royal Society of Chemistry.

**Figure 7 pharmaceutics-15-02174-f007:**
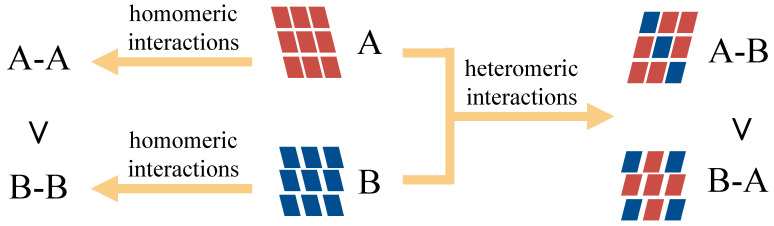
HBP approach for screening of multicomponent crystal forms.

**Figure 8 pharmaceutics-15-02174-f008:**
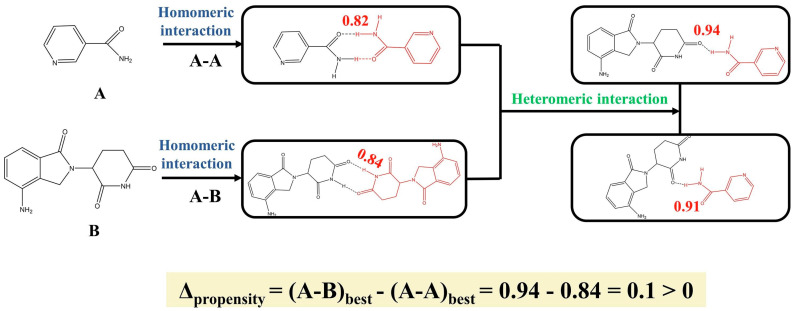
HBP calculation for the system lenalidomide and nicotinamide. Reprinted with permission from ref. [42]. Copyright 2021 Elsevier.

**Figure 9 pharmaceutics-15-02174-f009:**
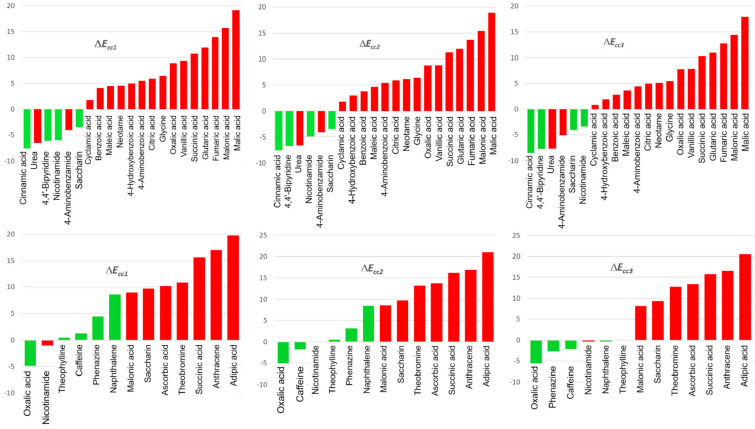
Co-formers of indomethacin (**top**) and paracetamol (**bottom**) co-crystals. Successful and failed cases are highlighted in green and red, respectively. Reprinted with permission from ref. [102]. Copyright 2020 American Chemical Society.

**Figure 10 pharmaceutics-15-02174-f010:**
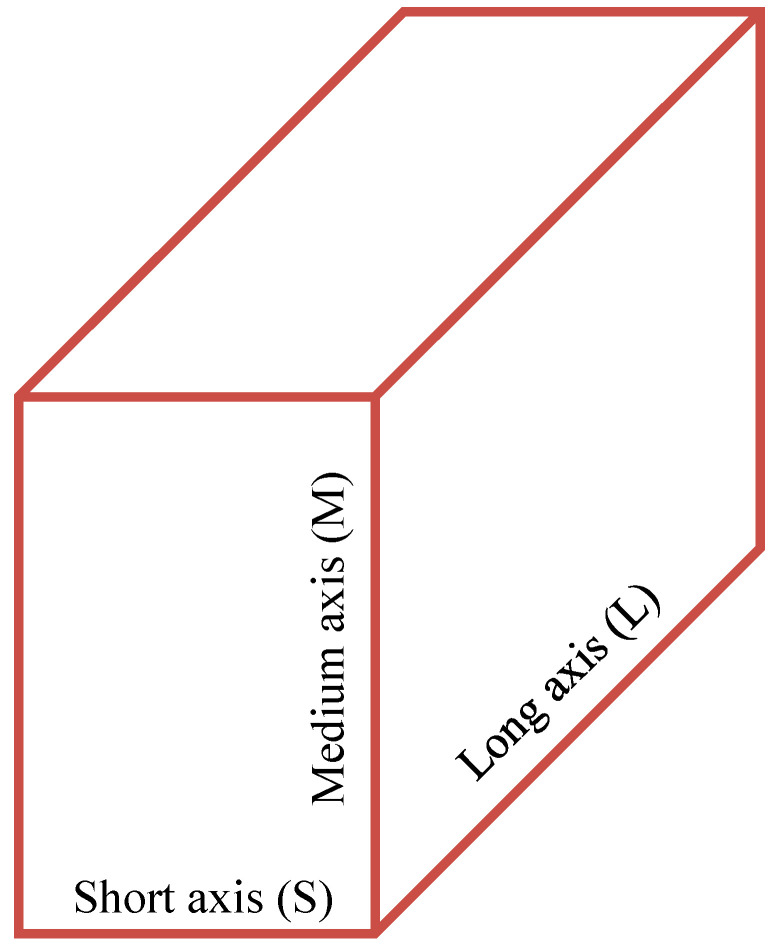
Three unequal dimensions of a model box.

**Figure 11 pharmaceutics-15-02174-f011:**
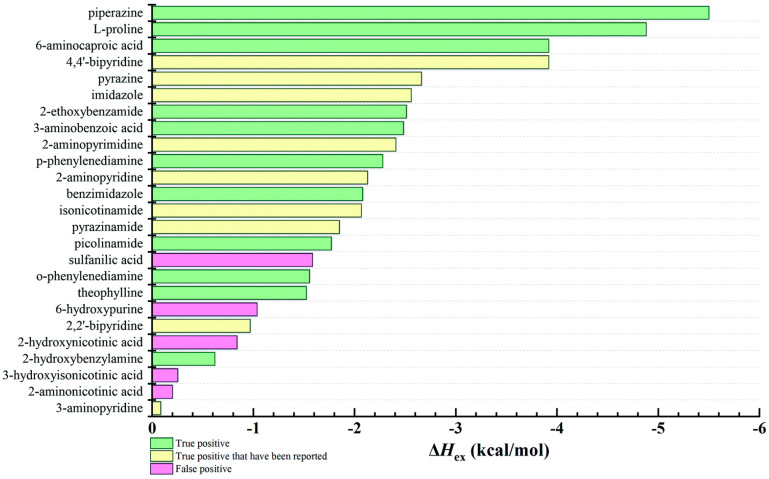
Multicomponent crystal forms screening of 2,4-dichlorophenoxyacetic acid based on COSMO-RS and MC methods. Reprinted with permission from ref. [106]. Copyright 2022 Royal Society of Chemistry.

**Figure 12 pharmaceutics-15-02174-f012:**
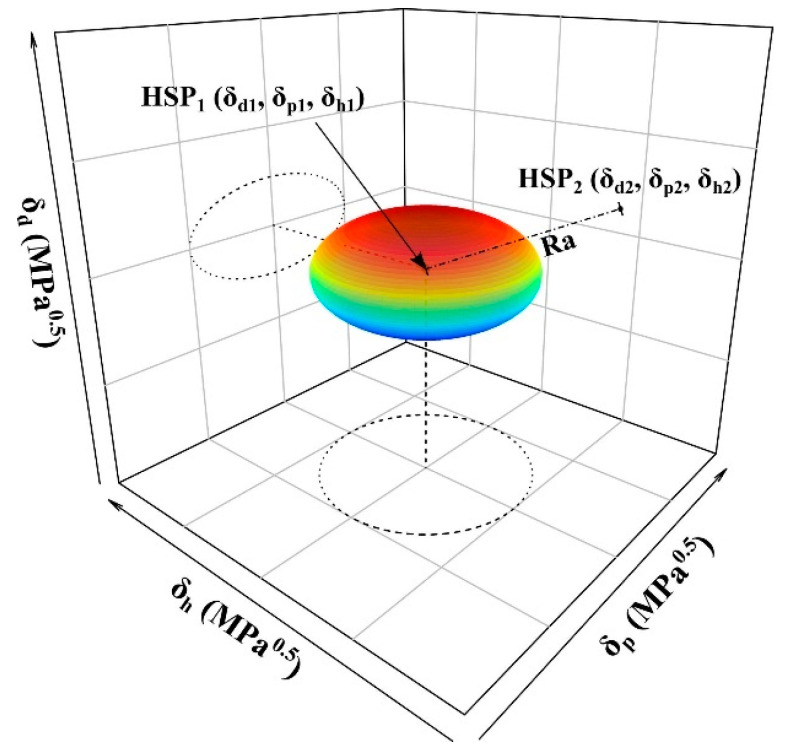
Hansen spheroid with Ra indicating the distance between HSP_1_ and HSP_2_. Reprinted with permission from ref. [120]. Copyright 2022 Elsevier.

**Figure 13 pharmaceutics-15-02174-f013:**
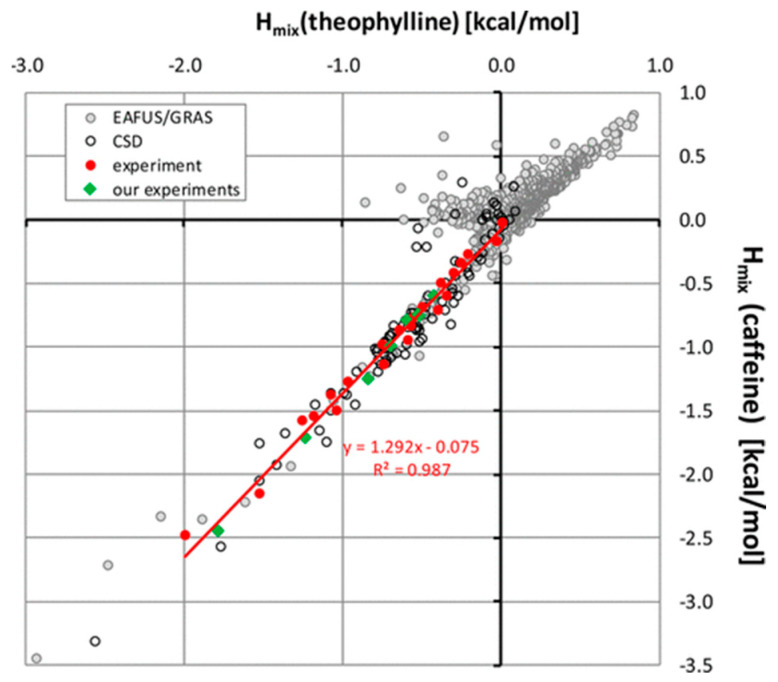
Relationship of *H*_mix_ between caffeine and theophylline. Reprinted with permission from ref. [52]. Copyright 2017 American Chemical Society.

**Figure 14 pharmaceutics-15-02174-f014:**
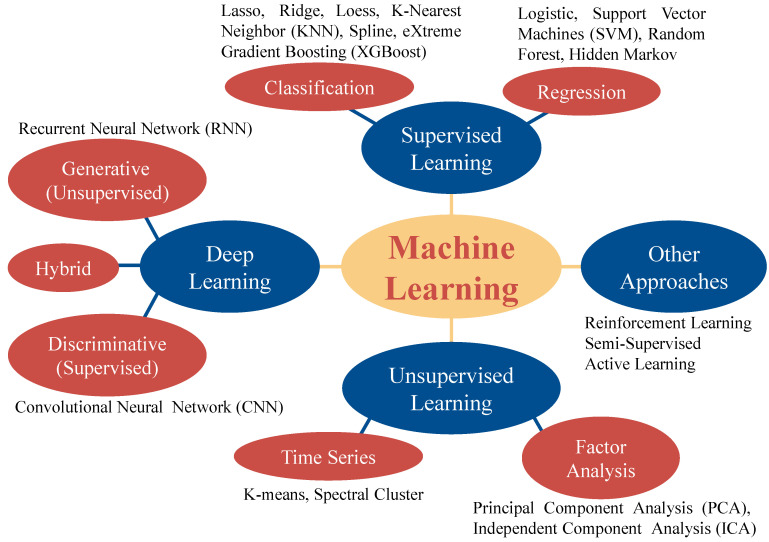
Commonly used ML algorithms.

**Figure 15 pharmaceutics-15-02174-f015:**
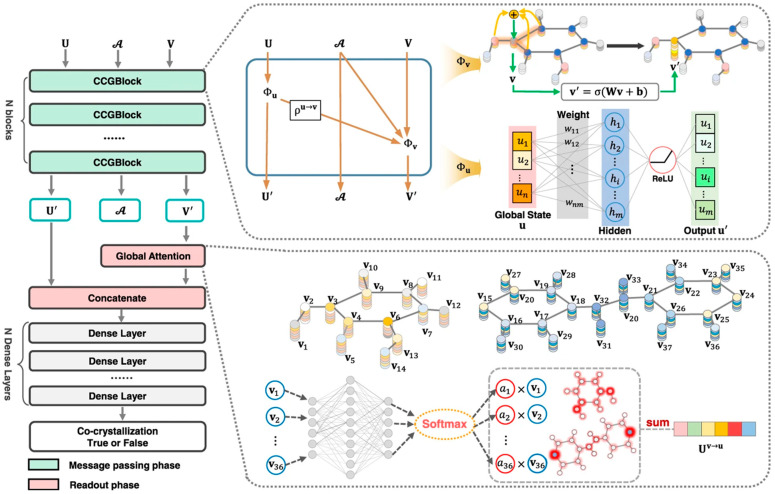
Screening framework of co-crystals. Reprinted with permission from ref. [130]. Copyright 2021 Springer Nature.

**Figure 16 pharmaceutics-15-02174-f016:**
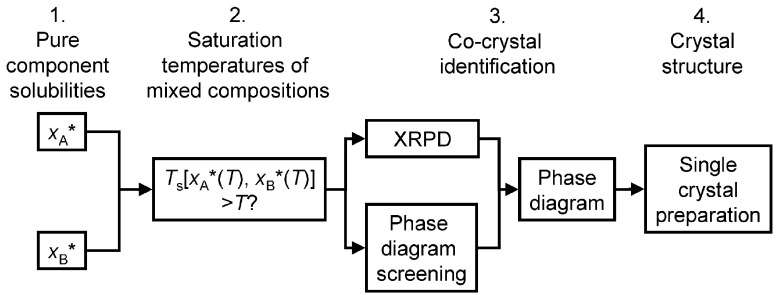
Steps for screening the formation of co-crystals based on the solubility of the pure components. Reprinted with permission from ref. [137]. *: at constant temperature. Copyright 2009 American Chemical Society.

**Table 1 pharmaceutics-15-02174-t001:** Hydrogen bond based methods for co-former screening and formation prediction of multicomponent solid forms.

Prediction Method	System	API	Co-Former	Preparation Method	Key Findings	Ref.
MEPs (only for multicomponent crystal forms)	multicomponent crystal forms	5,5′-di(pyridin-3-yl)-2,2′-bithiophene, 5,5′-di(pyridin-4-yl)-2,2′-bithiophene (T2)	7 aromatic and one aliphatic acid	liquid-assisted grinding	Among the 16 combinations, 8 single-crystal structures were obtained. MEPS calculations predicted synthon outcomes, matching experimental co-crystallization results except for T2:4-hydroxybenzoic acid.	[34]
	multicomponent crystal forms	kaempferol, quercetin, myricetin	praziquantel	suspension-stirring	Different conformations were analyzed to predict the possible interactions between API and co-formers, which were consistent with the interactions in 4 co-crystals obtained. Calculating the difference in interaction site pairing energies is more efficient for virtual screening of co-crystals.	[38]
	multicomponent crystal forms	spironolactone, griseofulvin	4-*tert*-butylpheno, phenol, 2,5-xylenol	liquid-assisted grinding	310 potential co-formers were screened based on the difference in the solid-state interaction site pairing energies (Δ*E*), 35 of these which showed the highest values of Δ*E* and were selected for experimental screening. 1 griseofulvin co-crystal and 2 spironolactone co-crystals were obtained.	[35]
	multicomponent crystal forms	resveratrol	4,4′-bipyridine, piperazine, phenazine, 1,10-phenanthroline, 1,4-diazabicyclo [2.2.2]octane, methenamine, acridine, succinimide, N,N-dimethyl-4-aminopyridine	solvent evaporation	10 new co-crystals of resveratrol with 9 co-formers were developed by analyzing interaction site pairing energy differences.	[36]
	multicomponent crystal forms	1,2,4-thiadiazole derivative (TDZH)	6 acids: oxalic, maleic, fumaric, adipic, vanillic and gallic	liquid-assisted grinding, solvent evaporation	5 new multi-component crystals of TDZH were discovered. Quantitative analysis of MEPs and periodic density functional theory calculations were used to examine the relationship between the donor/acceptor groups in TDZH/co-former molecules and the hydrogen bond pattern in multicomponent crystal forms.	[37]
supramolecular synthon (only for multicomponent crystal forms)	multicomponent crystal forms	5-fluorocytosine	sarcosine, dimethylglycine	liquid-assisted grinding	Co-crystal screening was guided using the relatively unexplored amine-carboxylate supramolecular synthon. 2 new co-crystals were discovered among 11 potential co-formers.	[29]
	multicomponent crystal forms	sulfadimethoxine	isonicotinamide, 4,4′-bipyridine, piperazine, 4,4′-trimethylenedipiperidine, 1,4-diazabicyclo [2.2.2]octane	liquid-assisted grinding, solvent evaporation	Co-formers containing cyclic amines, amides, carboxylic acids, and sulfonamide-based moieties were used to screen for sulfadimethoxine co-crystal formation. 8 new multicomponent crystal forms were obtained favoring amine derivatives.	[30]
	multicomponent crystal forms	regorafenib	malonic acid, glutaric acid, pimelic acid	liquid-assisted grinding	Regorafenib can form hydrogen bond supramolecular synthons with carboxylic groups. Dicarboxylic acids were chosen as candidates and 3 co-crystals were successfully formed.	[31]
	multicomponent crystal forms	nicotinamide, isonicotinamide	3,5-pyrazole dicarboxylic acid, dipicolinic acid, quinolinic acid	solvent evaporation	3 nitrogen heterocycle-containing aromatic dicarboxylic acids were chosen as successful co-formers based on carboxylic acid–aromatic N-hetero-synthons.	[32]
	multicomponent crystal forms	diclofenac	4,4′-bipyridine; 1,2-bis(4-pyridyl)ethane; 1,2-bis(4-pyridyl)propane; 2-aminopyridine, 3-aminopyridine, 4-aminopyridine, ethylenediamine	liquid-assisted grinding, solvent evaporation	Based on a robust acid···pyridine supramolecular synthon. Diclofenac acid formed 9 multicomponent crystal forms with pyridines and amine-based co-formers.	[33]
HBP (only for multicomponent crystal forms)	multicomponent crystal forms	pyrimethamine	carbamazepine, theophylline, aspirin, α-ketoglutaric acid, saccharin, p-coumaric acid	solvent evaporation	Hydrogen bond propensity calculations were conducted to predict co-crystallization results for pyrimethamine and 8 co-formers, resulting in the formation of 2 co-crystals and 4 salts of pyrimethamine.	[39]
	multicomponent crystal forms	pyrimethamine	oxalic acid, malonic acid, acetylenedicarboxylic acid, adipic acid, pimelic acid, suberic acid, azelaic acid	solvent evaporation	Hydrogen bond propensity calculations were carried out to predict the development of salts/co-crystals for pyrimethamine and 10 co-formers, accurately predicting 7 successful salts and 3 unsuccessful cases.	[40]
	multicomponent crystal forms	indomethacin	nicotinamide	milling	The hydrogen bond propensity demonstrates that the hydrogen bond motifs in indomethacin-nicotinamide co-crystal structures are one of the most likely donor-acceptor combinations.	[41]
	multicomponent crystal forms	lenalidomide	nicotinamide	solid-state/liquid-assisted grinding	Co-crystals were successfully synthesized based on the dominance of heteromeric interactions suggested by Δ_propensity_ (0.1) > 0.	[42]
Δp*K*_a_ rule (only for multi-component crystal forms)	multicomponent crystal forms	paracetamol	*trans*-1,4-diaminocyclohexane, 1,2-bis(4-pyridyl)ethane, 1,2-di(4-pyridyl)ethylene	liquid-assisted grinding, solvent evaporation	Δp*K*a < 0 were consistent with the co-crystal formation. Predicting proton transfer in solid-state based on Δp*K*a is not precise and can vary with the solvent used for crystallization.	[24]
	multi-component crystal forms	2-chloro-4-nitrobenzoic acid	2-bromopyridine, 3-amino-2-bromopyridine, 3-amino-2-chloropyridine, 2-amino-5-nitropyridine, 2-amino-3-bromopyridine, 2-chloro-3-hydroxypyridine, 2-amino-5-bromopyridine, 2-amino-5-chloropyridine, 2,6-dimethylpyridine	solvent evaporation	The results met well with the Δp*K*a rule: below 0, co-crystals are formed; above 3, salts are formed; and in the middle range, the formation of both is favored.	[25]
	multicomponent crystal forms	baicalein	dicotinamide	solvent evaporation, rotary evaporation, and cogrinding.	Co-crystal formation (Δp*K*a (−1.98)) between baicalein (p*K*a 5.4) and nicotinamide (p*K*a 3.4) was consistent with Δp*K*a rule.	[26]
	multicomponent crystal forms	AMG 517	10 acids: benzoic, *trans*-cinnamic, 2,5-dihydroxybenzoic, glutaric, glycolic, *trans*-2-hexanoic, 2-hydroxycaproic, l(+)-lactic, sorbic acid, l(+)-tartaric	slow cooling and solvent evaporation	Based on the Δp*K*a range (−2.33 to −4.12), 10 co-crystals of AMG 517 were formed.	[27]
	multicomponent crystal forms	3-hydroxybenzoic acid, 4-hydroxybenzoic acid, 6-hydroxy-2-naphthoic acid, 3-hydroxypyridine	pyrazine, 4-phenylpyridine, 1.2-bis(4-pyridyl)ethane, 4.4-bipyridine, quinoxaline, tetramethylpyrazine, *trans*-1,2-bis(4pyridyl)ethane, benzoic acid, isophthalic acid,	grinding, solvent-drop grinding, solvent-drop grinding	3 co-crystals (Δp*K*a < 0), 2 salts (Δp*K*a > 3), 8 co-crystals, 1 co-crystal of a salt and 1 solvated co-crystal (Δp*K*a 0.86 to 2.05) were successfully prepared, which were fully consistent with the Δp*K*a rule.	[28]

**Table 2 pharmaceutics-15-02174-t002:** Non-hydrogen bondbased methods for co-former screening and formation prediction of multicomponent solid forms.

Prediction Method	System	API	Co-former	Preparation Method	Key Findings	Ref.
HSP (for both multicomponent crystal forms and co-amorphous systems)	multicomponent crystal forms	indomethacin	nicotinamide, saccharin, 4,4′-bipyridine cinnamic acid	liquid-assisted grinding, reaction crystallization	Indomethacin was miscible with 21 out of 33 co-formers according to HSP results. Indomethacin formed co-crystals with 4 different co-formers, 2 of which were newly developed.	[48]
	multicomponent crystal forms	paliperidone	benzamide, nicotinamide, para hydroxy benzoic acid	solvent evaporation	Paliperidone formed co-crystals with benzamide, nicotinamide, and para hydroxy benzoic acid, as HSP predicted using three group contribution methods.	[49]
	co-amorphous	tadalafil	repaglinide	solvent evaporation	HSP difference between tadalafil (26.01 MPa^0.5^) and repaglinide (26.01 MPa^0.5^) is less than 7 MPa^0.5^, confirming their good miscibility. There were no molecular interactions in the system.	[50]
	co-amorphous	florfenicol	oxymatrine	solvent evaporation	The Δ*δ*_t_ value of florfenicol and oxymatrine is 3.87 MPa^0.5^, indicating miscibility.	[66]
	co-amorphous	norfloxacin	saccharin, naproxen, indomethacin, l-phenylalanine, l-arginine, l-tryptophan	dry ball mill	Δ$\bar{\delta }$ and Δ*δ*_t_ criteria were used to predict the molecular miscibility between norfloxacin and 17 co-former candidates Norfloxacin can form co-amorphous with 6 co-formers. Van Krevelen criterion is more suitable for assessing molecular miscibility in the formation of norfloxacin co-amorphous.	[67]
COSMO-RS (only for multicomponent crystal forms)	multicomponent crystal forms	carbamazepine	dl-mandelic acid, dl-tartaric acid, indomethacin	liquid-assisted grinding	21 out of 75 co-former candidates were investigated, where 3 new systems and 9 already known systems were obtained. Only using Δ*H*_ex_ as a criterion for selecting co-formers is not sufficient Considering both the fusion entropy Δ*S*_m_ and the excess enthalpy Δ*H*_ex_, the results have a better prediction.	[51]
	multicomponent crystal forms	caffeine, theophylline	8 phenolic acids	liquid-assisted grinding	8 new co-crystals were discovered on caffeine and theophylline. Caffeine and theophylline showed a linear correlation in the mixing enthalpies and co-crystal forming abilities.	[52]
	multicomponent crystal forms	posaconazole	4-aminobenzoic acid, l-malic acid, succinic acid, fumaric acid, ferulic acid, maleic acid, citric acid, l-hydroxy-2-naphthoic acid, gentisic acid, salicylic acid, l-lactic acid, adipic acid, 3,4-dihydroxybenzoic acid	high-throughput slurry, liquid-assisted grinding, reaction crystallization	COSMOquick was used to reduce a list of about 140 potential co-formers to 28 candidates. 13 new posaconazole co-crystals (7 anhydrous, 5 hydrates, and 1 solvate) were successfully prepared.	[53]
	multicomponent crystal forms	clotrimazole	3,5-dinitrosalicylic acid, 3,5-dinitrobenzoic acid indole-6-carboxylic acid, syringic acid, 3-nitrobenzoic acid, 1,4-naphthalenedicarboxylic acid, pyromellitic acid, 2,3-dihydroxybenzoic acid, 1,2,4-benzenetricarboxylic acid	liquid-assisted grinding, slurry suspension, solvent evaporation	14 out of 21 potential co-formers formed multi-component crystal forms with clotrimazole, including 5 reported cases, 3 new co-crystals and 6 new salts.	[54]
	multicomponent crystal forms	2-hydroxybenzylamine	succinic acid, p-aminobenzoic acid, p-nitrobenzoic acid, o-nitrobenzoic acid, p-toluic acid, 2,3-dihydroxybenzoic acid, 3,4-dihydroxybenzoic acid, p-nitrophenol, 5-hydroxyisophthalic acid	liquid-assisted grinding	21 out of 40 potential co-formers were characterized as new solid phases. 9 multicomponent single crystals were obtained and characterized.	[55]
MC (only for multi-component crystal forms)	multicomponent crystal forms	artemisinin	resorcinol, orcinol	liquid-assisted grinding	Only 2 out of 75 co-formers (3%) led to the formation of a co-crystal. Low success rate was due to artemisinin lacking strong hydrogen bond donors or acceptors.	[45]
	multicomponent crystal forms	leflunomide	pyrogallol, 3-hydroxybenzoic acid, 2-picolinic acid, 2-aminopyrimidine	liquid-assisted grinding	Structure and intermolecular interactions of leflunomide were analyzed using Isostar and Mercury to identify favorable functional groups for co-crystallization, resulting in 5 new co-crystals.	[46]
	co-amorphous and multi-component crystal forms	sulfamethoxazole	acetamide, propionamide, isonicotinamide, 2-hydroxypyridine, pyrazine, imidazole, oxalic acid dihydrate, N-hydroxysuccinimide, 1,2-di(4-pyridyl)ethylene, 1,2-di(4-pyridyl)ethylene, 1,3-di(4-pyridyl)propane, 4,4′-bipyridine, 4-phenylpyridine, benzamidine, carbamazepine, deoxycholic acid, hexamethylenetetramine, sodium deoxycholate	neat grinding, solvent evaporation	CSD motif search offered 39 potential candidates and MC was used to screen the compatibility. 13 new co-crystals, 1 salt and 4 co-amorphous systems were identified experimentally.	[47]
lattice energy (only for multicomponent crystal forms)	multicomponent crystal forms	carbamazepine	isonicotinamide	cooling crystallization, slurry	Carbamazepine-isonicotinamide formed co-crystals due to lower or comparable lattice energy of pure components. Carbamazepine-picolinamide could not form co-crystal due to less stable lattice energies than pure components.	[43]
	multi-component crystals	pentoxifylline	aspirin, salicylic acid, benzoic acid	neat and liquid-assisted grinding, solvent evaporation	The experimental and in silico screening of co-crystals yielded consistent results. FlexCryst software (www.flexcryst.com (accessed on 23 July 2023)) suggests that the prediction of co-crystal formation must satisfy ΔG ≥ −3 kJ/mol for feasibility, instead of ΔG ≥ 0 kJ/mol.	[44]
Artificial intelligence (for both multicomponent crystal forms and co-amorphous systems)	multicomponent crystal forms	diclofenac	iIsonicotinamide, 2-pyrrolidinone, 4,4′-Bipyridine.	neat grinding	An extreme gradient boosting model was developed using 1000 co-crystallization cases and 2083 chemical descriptors. 3 new co-crystals of diclofenac were discovered.	[56]
	multicomponent crystal forms	captopril	l-proline, sarcosine	liquid-assisted grinding	A random forest-based co-crystal prediction model was created using a dataset of positive samples from CSD and negative samples from randomly paired molecules. 2 captopril co-crystals verified the effectiveness of the model.	[57]
	multicomponent crystal forms	norfloxacin	nicotinamide, 4,4′-vinylenedipyridine	neat grinding	COSMO-SVM and 3D-CNN machine learning models were established. 2 new norfloxacin co-crystals were predicted and fabricated.	[58]
	co-amorphous	folic acid	nicotinamide, l-isoleucine, anthranilic acid, citric acid, theophylline, theobromine.	neat grinding	The gradient boost model achieved a predictive accuracy of over 73%. 6 novel co-amorphous forms of folic acid were predicted and discovered.	[59]
	co-amorphous	glycopyrronium bromide	budesonide, ethambutol	neat grinding	A molecular descriptor-based ML model was built with an accuracy of 79%. 2 successful co-amorphous and one failed case were confirmed by the model.	[60]
	multicomponent crystal forms	nimesulide	4,4′-bipyridine, trans-1,2-bis(4-pyridyl)ethylene, 1,2-bis(4-pyridyl)ethyne, 1,2-bis(4-pyridyl)ethane	liquid-assisted grinding, slurry	Co-crystal formation between nimesulide and pyridine analogues depends on various molecular descriptors of co-formers, with MEP having the greatest impact, followed by h_ema (sum of hydrogen bond acceptor strengths), Kier flex (molecular flexibility), and horizontal distance between two N atom projections.	[61]
	co-amorphous	carvedilol, mebendazole, carbamazepine, furosemide, indomethacin, simvastatin	20 natural amino acids	neat grinding	The PLS-DA model effectively separated co-amorphous and non-co-amorphous samples. Non-polar side chain amino acids were the most effective co-formers for the formation of co-amorphous systems, while polar amino acids were the least successful.	[62]

## Data Availability

No new data were created or analyzed in this study. Data sharing is not applicable to this article.

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
