# Peer review of "Recent Advances in Co-Former Screening and Formation Prediction of Multicomponent Solid Forms of Low Molecular Weight Drugs"

_pharmaceutics, 2023, doi:10.3390/pharmaceutics15092174_

Round 1

Reviewer 1 Report

Thank you to the authors for this comprehensive and useful review. An immense effort clearly went into the preparation of this work. We have some suggestions for improvement.

The title of the review was a bit misleading as this review was very highly focused on crystalline forms rather than amorphous formulations. You may wish to edit this to focus on what the review really covers.

To that point, and I know there is some difference of opinion in the literature, but the distinction between co-amorphous and amorphous solid dispersions put forth in the introduction is an artificial one. Both compose a single amorphous phase in which API and excipient are intimately mixed, but without long-range order. It is not necessary to give them different names just because one uses a polymer and one a small molecule. In fact, amorphous solid dispersions using non-polymers like cyclodextrin or trehalose are common, and standard nomenclature is to call them all amorphous solid dispersions (so long as they are molecularly dispersed)

In the supra-molecular synthons section, the discussion mainly reports the results from the studies. Instead, could the authors give more context to why supramolecular synthons have impacted the outcome? A more high-level summary would be more useful, elaborating what should be the expected outcome of the form screenings based on the likely interactions.

In the MEPs section: They haven't mentioned the level of calculations which are necessary for this work...ab initio? DFT, and what level? This would help to describe the amount of computational effort necessary to get good predictions without overkill or it taking too long.

During discussion of halogen bonds in the MEPs section, the authors can also talk about chalcogen and pnictogen bonds and how they are coming into more focus recently.

In Figure 8, should the cutoff for HBP always 0 in the authors' opinion? Or is it movable depending on the system (-0.1 or 0.1 for example) to get better fits to the experimental data?

A similar question for the COSMO-RS section...is the cutoff value dependent on the system or type of hydrogen bonds involved? Or should it always be the same?

Figure 15 should be eliminated as it is way too detailed for the scope of this review.

The AI system section seemed vague in how it provides value. Perhaps some discussion how AI systems could be used to take historical data from multiple prediction methods and produce a more accurate prediction might be a starting point.

I was hoping that the conclusions would include a venn diagram or comparison chart of the multiple methods' advantages and disadvantages. Also, discussion of how they overlap, and what methods might be complementary to produce a more accurate prediction. The purpose of a review paper is not only to summarize the current state of the literature, but also to synthesize that information at a higher level to help guide research efforts in the future. I think the authors have the expertise to do this if some extra time is taken, and could really help lots of researchers!

Thank you for the fine submission, and looking forward to reading the final version!

Reviewer 2 Report

Dear editor,

The review authored by Deng et al. highlights the historical progress of virtual screening for multicomponent systems of solids, such as cocrystals. The manuscript is well-written, and the coverage is sufficient for acceptance in your journal. However, I would also like to provide some suggestions to the authors. My point-to-point suggestions are listed below

1.      It is crucial to ensure that proper permissions are obtained before including pictures or figures from other manuscripts where the copyright still belongs to the original publisher. To avoid any copyright infringement, the authors should explicitly include a statement in the manuscript for each picture, clearly stating that the reuse of the picture has been permitted by the original publisher.

Here's a suggested statement that can be included for each picture:

"The image in this figure has been reproduced with permission from [Original Publisher]. Copyright for this image belongs to [Original Publisher]."

By providing such a statement for each picture, the authors can demonstrate that they have obtained the necessary permissions and are adhering to copyright regulations. This will also ensure that proper credit is given to the original publishers and authors of the images used in the review.

2.      I would recommend including the regulatory perspective definitions of salt and cocrystal in the introduction. At the end, the development of multicomponent systems is always geared towards gaining drug approval for the market, underscoring the significance of these definitions from a regulatory perspective, as outlined by organizations like the FDA or EMA.

3.      In Section 3.6, the coverage of other approaches for virtual cocrystal screening appears to be too superficial and lacks sufficient detail. It is recommended that the authors provide a more in-depth explanation of these approaches. For instance, the method proposed by ter Horst seems to resemble pure experimental screening, as inferred from Figure 16. Therefore, the authors should put in more effort to elaborate on the details of each approach in this section.

4.      I would recommend including a future outlook in the manuscript, discussing how virtual screening can be further improved and advanced.

One promising direction could be the integration and combination of various approaches mentioned in this review. For instance, combining machine learning algorithms with molecular docking methods could lead to more accurate predictions of cocrystal formations. Additionally, the utilization of quantum mechanical simulations in conjunction with virtual screening techniques may offer deeper insights into the thermodynamic stability and intermolecular interactions of multicomponent systems.

Furthermore, the application of high-throughput virtual screening methods and advancements in computing power could significantly speed up the identification of potential cocrystals. This could be further complemented by the incorporation of experimental data to validate the predictions made by virtual screening methods. Additionally, collaborations between computational scientists and experimentalists could foster a more synergistic approach to cocrystal discovery. By exchanging knowledge and expertise, researchers can benefit from the strengths of both computational and experimental methods, leading to more efficient and reliable cocrystal screening processes.

The future of virtual screening for multicomponent systems, such as cocrystals, holds great promise. By exploring novel combinations of approaches, leveraging technological advancements, and fostering collaborative efforts, we can anticipate significant advancements in the field, paving the way for the expedited discovery and development of innovative drug formulations

Reviewer 3 Report

The authors have addressed very important aspects of the low-molecular weighted drugs. In this review, the authors have presented an all-encompassing summary of current screening and prediction techniques utilized in the development of API multicomponent solid forms, encompassing both crystalline states (co-crystals and salts) and amorphous forms (co-amorphous). Moreover, it delves into recent progress and upcoming trends in prediction methods, with a specific emphasis on artificial intelligence. However, I have following minor comments:

1. Please realign the figures alignment throughout the manuscript.

2.Figure 11 is not clear. Please replace the image with the clear one.

3. The alignment of the equations are not uniform.

4. The content of the Figure 16 is not clear.

1. Minor punctuation and grammatical errors need to be resolved.
